# Time to sputum culture conversion, treatment outcomes, and associated factors among rifampicin-resistant or multidrug-resistant tuberculosis patients in the Sidama region, Ethiopia: A retrospective follow-up study

**Wolde Abreham Geda** [1]*, **Kebede Tefera Betru**[2], **Tarekegn Solomon**[2], **Solomon Daniel**[2]

**1** Sidama Regional Public Health Institute, Hawassa, Sidama, Ethiopia, **2** School of Public Health, College of Medicine and Health Sciences, Hawassa University, Hawassa, Sidama, Ethiopia

* wolde.abreham@yahoo.com

## Abstract

### Background

Time to sputum culture conversion is critical for follow-up treatment effectiveness in multidrug-resistant tuberculosis patients. However, the evidence regarding time to culture conversion, treatment outcomes, and the associated factors was sparse and inconsistent.

### Objective

This study aimed to determine the time to culture conversion, treatment outcomes, and associated factors among rifampicin-resistant and multidrug-resistant tuberculosis cases in the Sidama region from April 1 to December 31, 2024.

### Methods

We conducted a retrospective follow-up study of 346 patients who enrolled between January 2013 and June 2024. We collected data from patients' medical records using a standardized form, entered the data, and analyzed it using Stata 16.1 software. We performed the analysis using the Kaplan-Meier model for time to culture conversion, Weibull distribution gamma frailty, and logistic regression models to assess factors associated with time to culture conversion and treatment outcomes, respectively. We considered an adjusted hazard or odds ratio with a 95% CI and a p-value < 0.05 to determine significance.

### Results

Among the participants, 302 (87.3%) achieved culture conversion in a median time of 76 days (95% CI: 71–79 days). Patients with a history of previous loss to follow-up

**Data availability statement:** The entire de-identified data underling the study findings are available at DOI:10.6084/m9.figshare.30655226.

**Funding:** The author(s) received no specific funding for this work.

**Competing interests:** The authors have declared that no competing interests exist.

experienced an approximately fivefold delay in culture conversion (AHR = 0.2; 95% CI: 0.1–0.6; p = 0.001), while relapse cases had a twofold delay (AHR = 0.5; 95% CI: 0.2–0.9; p = 0.02) compared with new patients. The treatment success rate was 234/346 (67.6%). Female patients had higher odds of achieving a favorable treatment outcome (AOR = 1.8; 95% CI: 1.0–3.3; p = 0.04), whereas patients who experienced culture reversion had significantly lower odds of a favorable outcome (AOR = 0.05; 95% CI: 0–0.4; p = 0.001).

## Conclusions

The majority of patients experienced culture conversion within three months. However, patients with a history of loss to follow-up and relapse experienced delayed culture conversion. These findings highlight the urgent need to improve patient adherence.

## Introduction

Tuberculosis (TB) is the deadliest infectious disease, but it is curable if patients get the appropriate treatment regimen. Roughly, TB infects a quarter of the global population. However, most infected individuals will not develop active TB disease, and some individuals may recover due to their immune response [1].

Globally, approximately 10.7 million people were diagnosed with TB in 2024 [2]. Among people who were diagnosed with TB, 390,000 people additionally acquired multidrug-resistant/rifampicin mono-resistant-TB (MDR/RR-TB). Ethiopia is one of the 30 countries with a high burden of TB and human immunodeficiency virus (HIV) associated TB. In 2024, the estimated annual incidence of TB was 141 cases per 100,000 people, accompanied by a death rate of 16 per 100,000. Although Ethiopia has transitioned out of the list of countries with high MDR-TB burden since 2020, MDR-TB continues to pose a significant public health challenge. In 2024 alone, an estimated 2500 new cases of RR-TB occurred, with a considerable number of cases remaining undetected [1].

The resistance to rifampicin, which is the most effective first-line anti-TB treatment, has posed a remarkable concern. When resistance occurs to both rifampicin and isoniazid, the condition is known as multidrug-resistant TB. The majority of patients with RR-TB also exhibit resistance to isoniazid, thereby fulfilling the criteria for MDR-TB. Therefore, both MDR-TB and RR-TB are treated with second-line medications [2, 3].

Over the years, global and local stakeholders have made vast efforts to prevent and control transmission, reduce mortality, and mitigate the associated disease burden. The global campaigns undertaken to combat tuberculosis include directly observed short-course treatment and the current End-TB strategy, among others. However, TB remains one of the public health challenges, especially in developing countries [2].

Proper diagnosis and treatment follow-up are needed to ensure better treatment success rate and halt the emergence of MDR-TB or extensively drug-resistant (XDR) TB [4].

The second-line anti-TB treatments are weaker in clearing the bacteria while causing greater toxic side effects to the patients. Treatment adherence and close monitoring are crucial in maximizing the favorable treatment outcomes. Sputum culture conversion is a surrogate indicator of treatment effectiveness [4].

It is a tool to determine the duration of treatment for MDR-TB patients. During the intensive phase, the MDR-TB treatment continued for at least four months after the culture conversion or eight months, whichever is longer [5].

The previous study reported a median time to sputum culture conversion of 65 days, with a cumulative probability of 0.89 at the end of the fourth month. The distribution of treatment outcomes was as follows: cured- 216 (55.1%), treatment completed- 12 (3.1%), died- 44 (11%), lost to follow-up- 28 (7%), on treatment- 80 (20%), and others 16 (4%) [4].

Another study reported a median time to sputum culture conversion of 85 days. In addition, the study found that older age was significantly associated with delayed culture conversion and poorer treatment outcomes [5].

Although the World Health Organization (WHO) has introduced shorter treatment regimens such as BPaLM (bedaquiline, pretomanid, linezolid, and moxifloxacin), most MDR/RR-TB regimens still last 18–20 months or longer.

Therefore, periodic culture evaluations are essential for tracking treatment progress, planning and implementing respiratory isolation, determining treatment duration, and modifying regimens asnecessary [6].

Moreover, understanding factors associated with time to culture conversion and treatment outcomes helps to improve patient outcomes by altering their impact [7].

## Methods

### Study area

This study took place in the Sidama Region of Ethiopia. The region is located in the southern part of the country. The region borders the Oromia Region to the north, northeast, south, and southeast. The South Ethiopia region borders to the south. Geographically, the Sidama region lies between 6°14′ and 7°15′ North latitude and 37°9′ to 39°14′ East longitude.

According to the 2007 Central Statistical Agency (CSA) census, 4,647,672 people lived in the Sidama region. The region's healthcare system includes 22 hospitals, 142 health centers, and 553 health posts. Among these hospitals, only the Yirgalem hospital provided treatment for RR-TB and MDR-TB, while other hospitals and health centers provide follow-up care.

### Study design and period

We conducted a retrospective follow-up study on MDR/RR-TB patients at the Yirgalem treatment initiation center in the Sidama region of Ethiopia. We collected the de-identified patient data, using the patients' medical and MDR-TB record numbers, from April 1 to December 31, 2024.

### Source and study population

The source population included all tuberculosis patients in the Sidama Region. The study population consisted of patients with RR/MDR-TB who were registered at the Yirgalem treatment initiation center between January 2013 and June 2024.

### Inclusion and exclusion criteria

We included RR/MDR-TB patients who were 15 years of age or older and had a baseline positive culture result at the time of treatment initiation.

We excluded patients with incomplete medical records and those diagnosed with RR/MDR-TB but who died before the initiation of anti-TB treatment.

The total eligible RR/MDR-TB population in the region during the study period was 420. Among these, 74 patients did not meet the inclusion criteria, leaving a final sample size of 346.

## Data collection

The researchers followed the study participants for 29,209 person-days. Data were collected from patients' medical records using a standardized data extraction form. The information obtained included demographic characteristics (sex and age), clinical variables (time to sputum culture conversion, treatment outcomes, and treatment regimens), and associated factors (HIV status and previous treatment history). They entered the collected data into Microsoft Excel and imported it into Stata 16.1 software for statistical analysis.

## Data integrity

We initially reviewed 5% of patient medical records to assess how well the data collection tool functioned. After making minor adjustments based on this assessment, we proceeded with the full-scale data collection. We recruited data collectors from among health professionals with relevant clinical backgrounds and provided them with a two-day training session on the use of the data extraction tool. Each day, our team reviewed the gathered information for completeness before leaving the facility. We cleaned the dataset by performing frequency analyses in Stata 16.1 version to verify variable consistency, detect missing data, and evaluate their validity. To ensure data integrity, we randomly rechecked 5% of the collected records.

## Dependent and independent variables

The dependent variables in this study include time to sputum culture conversion and treatment outcomes. Treatment outcomes were cure, treatment completion, death, loss to follow-up, treatment failure, not evaluated, and culture reversion.

The independent variables considered in relation to time to culture conversion and treatment outcomes include several demographic, clinical, and patient-related factors. Demographic variables were sex and age categories. Clinical variables comprise the type of drug resistance, specifically rifampicin mono-resistance and multidrug resistance. Patient-related factors include HIV status and patient registration groups. Treatment-related factors encompass treatment regimen, categorized as short-term or long-term therapy.

## Definition of the outcome variables

Time to sputum culture conversion: the duration between the time of treatment initiation and the date of the first of the two consequent culture conversions, collected at least 30 days apart. It refers to the time required for an event to occur in this study [6].

Cured: a confirmed pulmonary TB case at the time of treatment initiation, who completed the treatment recommended by the national guideline, with evidence of bacteriological response and without evidence of failure.

Treatment completed: a patient who completed treatment recommended by the national policy, but the outcome does not fulfill the criteria of cure or treatment failure.

Treatment failed: a pulmonary TB patient whose treatment regimen needed to be stopped or changed permanently.

Died: a TB patient who died before the start of treatment or during the treatment course

Loss to follow-up: a TB patient who did not start treatment or interrupted treatment for two consecutive months or more.

Not evaluated: a TB patient with no treatment outcome judgment.

Treatment success: the sum of patients who were cured and completed the treatment regimen [8].

Culture reversion: time to culture reversion is the time between the date of the second negative culture marking culture conversion and the first of the two positive cultures that indicate reversion [9].

## Definitions of the treatment regimens

Long-term regimens are an extended treatment course for RR/MDR-TB. The treatment regimen typically lasts 18 months or longer. Depending on the patient's drug resistance profile, previous treatment history, and tolerance to medications, the regimen can follow either a standardized or an individualized approach.

The standardized treatment: In this approach, all eligible patients receive uniform treatment.

The individualized treatment: in contrast, the eligible patients get treatment in a tailored manner, based on their drug susceptibility testing profile.

Short-term regimens are RR/MDR-TB treatment regimens that span 9–12 months; the BPaLM regimen, given for six months, is an all-oral treatment regimen. This group of regimens lasts shorter, compared to the long-term treatment regimens that extend 18 months or above [8].

## Statistical analysis

The researchers completed the standardized form, entered it into Excel, and imported the data into Stata version 16.1 for statistical analysis. We performed descriptive summary statistics, frequencies, percentages, means, medians, and standard deviations to summarize the socio-demographic and clinical variables.

After declaring the survival time data, the research team conducted the Schoenfeld residual test to evaluate whether the data fulfilled the proportional hazard assumption. Consequently, the team conducted model selection for survival analysis, employing the Akaike information criterion (AIC). They run the Weibull Gamma Frailty model for bivariable and multivariable analysis. The team assessed the model's goodness of fit by employing the Cox-Snell residual test.

They performed Kaplan-Meier survival analysis to estimate the time until sputum culture conversion and the cumulative probabilities of remaining culture-positive at various time points. They fitted a parametric frailty model and conducted bivariable and multivariable analysis to identify the potential factors associated with time to sputum culture conversion.

The research team conducted a binary logistic regression analysis to identify factors associated with unfavorable treatment outcomes and treatment abandonment. The team selected potential predictor variables based on prior research and their clinical relevance. We constructed nested logistic regression models sequentially by adding one predictor at a time. The team evaluated the fit of each model using likelihood ratio (LR) tests and the Akaike Information Criterion (AIC) to determine the most parsimonious model. The team selected the predictors that improved the model fit or reduced the AIC in the final model.

Finally, the team identified the strength of the association using the adjusted hazard ratio (AHR) and adjusted odds ratio (AOR) for factors associated with conversion time, along with the corresponding 95% Confidence Interval (CI) and a significance level of $p < 0.05$.

## Ethical considerations and participant agreement

We obtained ethical approval for this study from Hawassa University's Institutional Review Board (IRB) and a support letter from the Sidama Regional Public Health Institute. We submitted the ethical clearance documents to Yirgalem Hospital. Because the study was retrospective, we could not obtain individual patient consent. Instead, we obtained written informed consent from the director of the Yirgalem MDR-TB treatment initiation center to access and extract data from medical records. We also discussed the study's purpose and benefits with the treatment center staff to ensure transparency. We de-identified all data, stored them securely in locked cabinets, and protected the database with passwords to maintain patient confidentiality. We conducted the study in accordance with the 1964 Declaration of Helsinki.

## Results

A total of 420 patients with RR or MDR-TB were enrolled over an 11-year follow-up period. Among the patients enrolled, 74 were excluded for not meeting the criteria, 55 had incomplete records, 18 were under age fifteen, and 1 patient had extra-pulmonary TB. Finally, the study analyzed 346 patients. Among the 346 participants, 225 (65%) were male and 121

(35%) were female. The mean and median ages of the participants were 30 and 27 years, respectively, with a standard deviation of ±11.78 (Table 1).

## Time to sputum culture conversion

Among the 346 study participants, 302 (87.3%) achieved sputum-culture conversion in a median time of 76 days (95% CI: 71−79 days). Additionally, 36 patients (10.4%) displayed a delayed conversion, with a median conversion time of 141 days (95% CI: 132−151 days). On the other hand, 8 (2.3%) patients did not show culture conversion by the end of the treatment (95% CI: 0.023, 0.01–0.04, p<0.05). The follow-up period for participants totaled 29,209 person-days. The cumulative probabilities of survival or sputum culture non-conversion at the end of the second, third, fourth, and sixth months were 0.90, 0.55, 0.22, and 0.15, respectively. (Figs 1 and 2).

## Factors associated with time to sputum culture conversion

The researchers conducted a model comparison for various survival models. They determined that the Weibull-Gamma frailty model provided the best fit for the data, with an AIC of 600.7. They observed a highly significant frailty variance, $\theta = 1.0$, with a p-value of less than 0.001, as detailed below (Table 2).

**Table 1. Baseline features of the patients at the time of initiation of second-line anti-TB treatment in the Sidama region, Ethiopia, from January 1, 2013 to June 30, 2024 (n = 346).**

| Variables | Frequency | Percent |
|---|---|---|
| Age category | | |
| 15-24 | 109 | 31.5 |
| 25-34 | 135 | 39.0 |
| 35-44 | 51 | 14.7 |
| 45-54 | 32 | 9.3 |
| >55 | 19 | 5.5 |
| Total | 346 | 100.0 |
| HIV status | | |
| HIV-positive | 31 | 9.0 |
| HIV-negative | 315 | 91.0 |
| Total | 346 | 100.0 |
| Resistance type | | |
| RR-TB | 329 | 95.0 |
| MDR-TB | 17 | 5.0 |
| Total | 346 | 100.0 |
| Patient registration group | | |
| New | 65 | 18.8 |
| Relapse | 160 | 46.2 |
| Loss to follow-up | 51 | 14.7 |
| Treatment failure | 70 | 20.2 |
| Total | 346 | 100.0 |
| Treatment regimen | | |
| Short-term treatment | 76 | 22.0 |
| Long-term treatment | 270 | 78.0 |

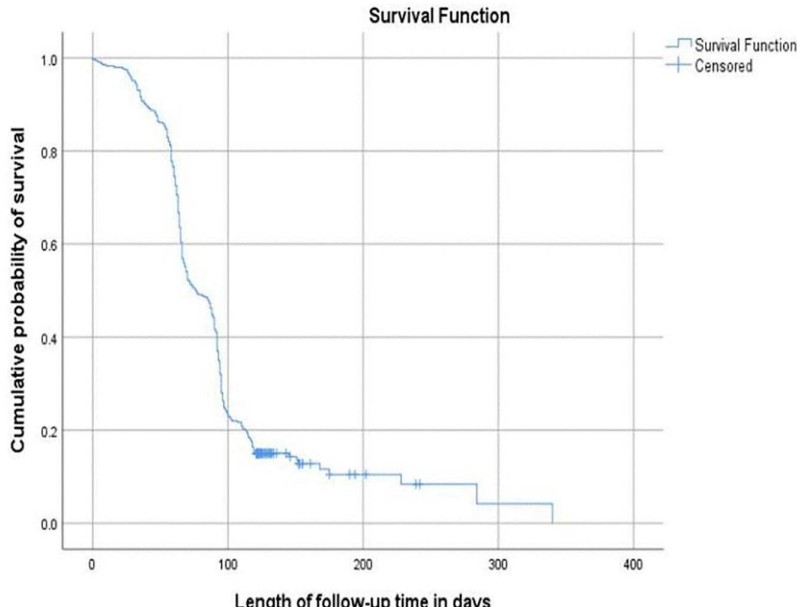

**Fig 1. Kaplan-Meier graph of time to sputum culture conversion among rifampicin/multidrug-resistant tuberculosis patients in the Sidama region, from January 1, 2013 to June 30, 2024.**

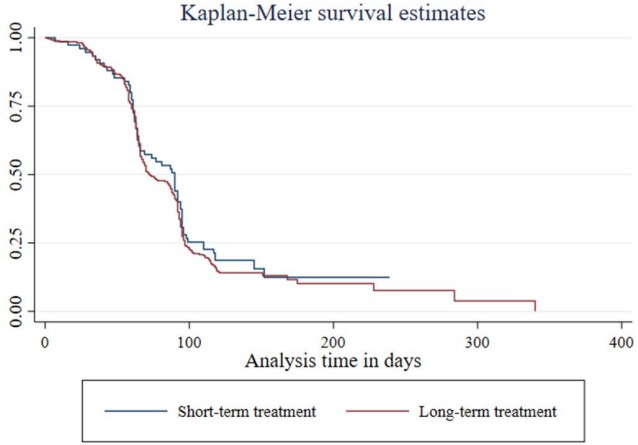

**Fig 2. Kaplan-Meier graph of time to culture conversion versus treatment regimens, among rifampicin/multidrug-resistant tuberculosis patients in the Sidama region, from January 1, 2013, to June 30, 2024.**

## The Schoenfeld residuals test to evaluate the proportional hazard assumption

We evaluated the proportional hazards assumption using Schoenfeld residuals. The plot of the scaled Schoenfeld residuals against time showed that the smoothed residual lines remained approximately horizontal, indicating that the hazard ratios stayed constant over time. This graphical assessment, therefore, suggested no evidence of violation of the proportional hazards assumption (Fig 3).

**Table 2. Survival analysis models comparison using Akaike information criteria and Log likelihood for time-to-sputum culture conversion and the associated factors.**

| Model | Baseline hazard | Frailty | Variance | Log likelihood | Akaike information Criteria |
|---|---|---|---|---|---|
| Cox proportional hazard | | | | −1239.2 | 2480.0 |
| Weibull regression | Weibull | | | −324.0 | 652.7 |
| Gompertz regression | Gompertz | | | −370.0 | 743.8 |
| Exponential regression | Exponential | | | −406.0 | 818.7 |
| Frailty distribution | Weibull | Inverse Gaussian | 0.84 (P<0.001) | −309.6 | 625.0 |
| Frailty distribution | Weibull | Gamma | 1.00 (P<0.001) | −297.0 | 600.7 |

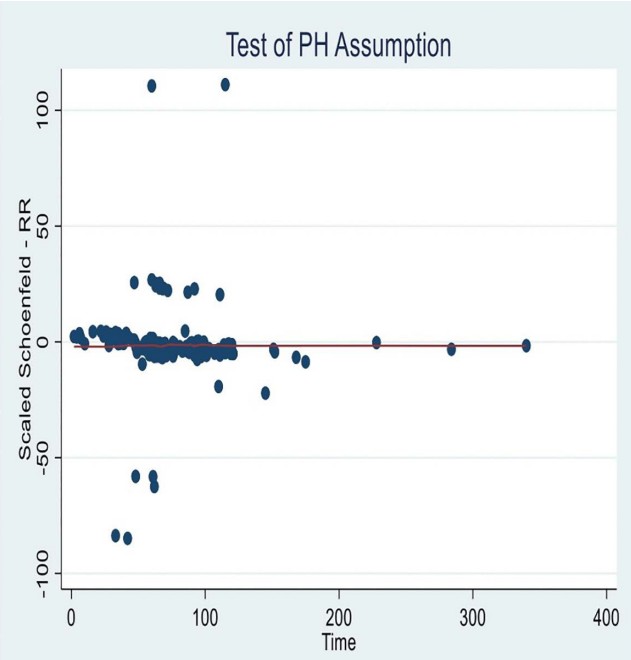

**Fig 3. Schoenfeld residuals tests to check proportional hazard assumption.**

### The cox snell test of best fit for the weibull gamma frailty model

We used the Cox-Snell residuals to assess the goodness-of-fit of the Weibull Gamma Frailty model. The cumulative hazard of the residuals is approximately aligned with the 45° diagonal line, indicating that the model fits the data well (Fig 4).

In multivariable analysis, compared with a new patient registration group, patients who experienced a loss to follow-up had a lower hazard of sputum culture conversion (AHR=0.20; 95% CI: 0.10–0.60; p=0.001). This indicates that patients lost to follow-up had approximately a fivefold longer time to culture conversion compared with new patients. In addition, patients in the relapse registration group had a 50% lower hazard of sputum culture conversion compared with new patients (AHR=0.50; 95% CI: 0.20–0.60; p=0.02), indicating that relapse patients experienced nearly a twofold delay in culture conversion compared with new patients.

On the other hand, there was no significant association between study participants' age category, sex, HIV status, resistance type, and MDR-TB treatment regimen and the culture conversion (Table 3).

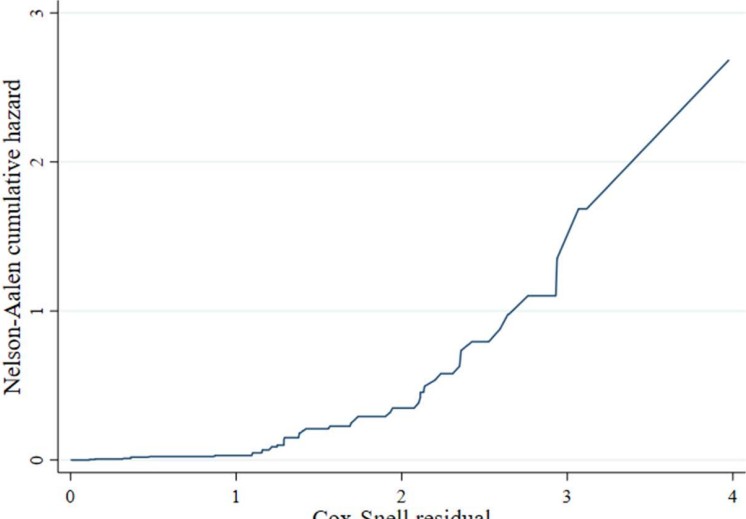

**Fig 4. Cox Snell residual test of goodness of fit for Weibull Gamma Frailty Model.**

## Treatment outcomes

Among the 346 study participants, 32 patients (9.2%) were not evaluated for their treatment outcomes. Of these, 26 patients had not completed their treatment course, while 6 patients had shifted to other facilities.

The overall treatment success rate was 234 (67.6%). The cure and treatment completion rates were 177 (51.2%) and 57 (16.5%), respectively. In contrast, the rate of unfavorable treatment outcomes was 80 (23%). Among these, 23 (6.6%) died, 5 (1.4%) experienced treatment failure, and 52 (15%) cases were lost to follow-up (Fig 5).

## Factors associated with treatment outcomes

We conducted model selection using the AIC. Although the models exhibited relatively high AIC values, we chose to include all the variables examined in previous studies. This approach allowed us to explore their potential associations with unfavorable treatment outcomes, ensuring a comprehensive analysis and maintaining comparability with the existing literature (Table 4).

A sex group of females had a higher probability of a favorable treatment outcome (AOR = 1.8; 95% CI: 1.0–3.3; p = 0.04).

A patient who experienced culture reversion had a lower odds of a favorable treatment outcome (AOR = 0.05, 95% CI: 0–0.4; p = 0.001).

On the other hand, different age categories, HIV status, drug resistance types, patient registration groups, and treatment regimens were not significantly associated with unfavorable treatment outcomes (Table 5).

## Treatment abandonment

We ran a stepwise model selection analysis to assess the potential association between patient registration groups and the current treatment abandonment. As a result, we determined that Model 1 was the best-fitting and most parsimonious model compared to the others, with an AIC of 254 (Table 6).

In the bivariable analysis, patients in the treatment failure registration group demonstrated a marginal association with treatment abandonment, showing a crude odds ratio (COR) of 0.3 (95% CI: 0.1–1.0, p = 0.04). However, there was no significant association between the patient registration groups and treatment abandonment (Table 7).

**Table 3. Factors associated with the delayed time to sputum culture conversion among rifampicin/multidrug-resistant tuberculosis patients in the Sidama region from January 1, 2013 to June 30, 2024 (n = 346).**

| Variables (n = 346) | N (%) | Culture conversion N (%) | | CHR(95% CI) | p-value | AHR(95% CI) | p-value |
|---|---|---|---|---|---|---|---|
| Sex | Total N = 346 | Event N = 302 | Censored = 44 | | | | |
| Male | 225 (65) | 195 (86.7) | 30 (13.3) | Reference | | Reference | |
| Female | 121(35) | 107 (88.4) | 14 (11.6) | 1.2 (0.8-1.9) | 0.360 | 0.8 (0.5- 1.3) | 0.360 |
| Age | | | | | | | |
| 15 −24 | 109 (31.5) | 89 (81.7) | 20 (18.3) | Reference | | Reference | |
| 25-34 | 135 (39) | 124 (92) | 11 (8) | 0.9 (0.5-1.4) | 0.510 | 0.8 (1-1.4) | 0.460 |
| 35-44 | 51 (14.7) | 47 (92.2) | 4 (7.8) | 1.0 (0.5-1.8) | 0.890 | 0.9 (0.4-1.7) | 0.680 |
| 45-54 | 32 (9.2) | 28(87.5) | 4 (12.5) | 1.0 (0.5-2.2) | 0.940 | 0.9 (0.4-2) | 0.810 |
| >55 | 19 (5.5) | 14 (74) | 5 (26) | 1.5 (0.6-3.9) | 0.390 | 1.3 (0.5-3.5) | 0.580 |
| HIV status | | | | | | | |
| HIV-positive | 31 (9) | 26 (84) | 5 (16) | Reference | | Reference | |
| HIV-negative | 315 (91) | 276 (87.6) | 39 (12.4) | 1.0 (0.5-2.1) | 0.930 | 1.0 (0.5-2.1) | 0.930 |
| Drug-resistance type | | | | | | | |
| RR-TB | 329 (95) | 286 (87) | 43 (13) | Reference | | Reference | |
| MDR-TB | 17 (5) | 16 (94) | 1 (6) | 0.5 (0.2-1.3) | 0.150 | 0.5 (0.2-1.3) | 0.150 |
| Registration group | | | | | | | |
| New | 65 (19) | 52 (80) | 13 (20) | Reference | | Reference | |
| Treatment failure | 70 (20) | 61 (87) | 9 (13) | 1.7 (1.0-3.0) | 0.070 | 0.8 (0.4-1.7) | 0.600 |
| Relapse | 160 (46) | 141 (88) | 19 (12) | 0.7 (0.5-1.2) | 0.190 | 0.5 (0.2-0.9) | 0.020 |
| Loss to follow-up | 51 (15) | 48 (94) | 3 (6) | 0.4 (0.2-0.9) | 0.010 | 0.2 (0.1- 0.6) | 0.001 |
| MDR-TB treatment | | | | | | | |
| Short-term | 76 (22) | 64 (84) | 12 (16) | Reference | | Reference | |
| Long-term | 270 (78) | 238 (88) | 32 (12) | 0.8 (0.5-1.3) | 0.35 | 0.8 (0.5-1.3) | 0.350 |

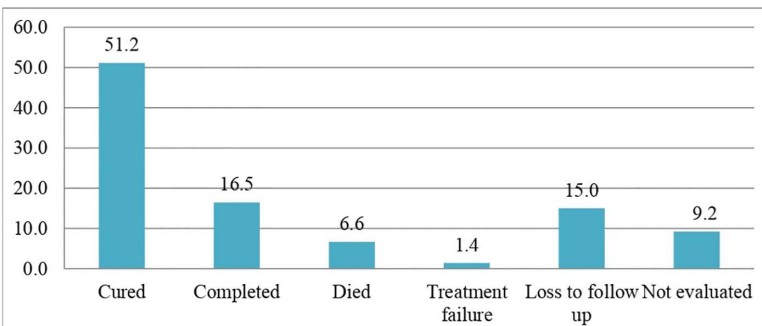

**Fig 5. Treatment outcomes in rifampicin/multidrug-resistant tuberculosis patients in the Sidama region, from January 1, 2013, to June 30, 2024.**

## Discussions

Some previous studies have reported an initial median time to sputum culture conversion among RR/MDR-TB patients of less than or equal to two months [10–15]. On the other hand, the current study revealed a median time of 76 days, which was consistent with other study findings that reported the median time of approximately two to three months [4–6,16–24].

**Table 4. Logistic regression model selection for unfavorable treatment outcome for predictor variables among rifampicin/ multidrug-resistant tuberculosis patients.**

| Model | Predictors | Log likelihood | LR X² (df) | p. value | Pseudo (R²) | AIC |
|---|---|---|---|---|---|---|
| Model 1 | Sex | −177.3 | 1.90 | 0.168 | 0.001 | 358 |
| Model 2 | Sex + HIV infection | −177.0 | 2.03 | 0.362 | 0.001 | 360 |
| Model 3 | Sex + HIV infection+ MDR-TB treatment | −177.0 | 2.20 | 0.533 | 0.006 | 362 |
| Model 4 | Sex + HIV infection+ MDR-TB treatment+ registration group | −177.0 | 2.38 | 0.666 | 0.007 | 364 |

**Table 5. Factors Associated with unfavorable treatment outcomes among the study participants (n = 346). N.B.: 32 patients were not evaluated for their treatment outcomes; therefore, treatment success was assessed in 314 patients.**

| Variables (n = 346) | N (%) | Treatment outcome N (%) | | COR | p-value | AOR | p-value |
|---|---|---|---|---|---|---|---|
| Sex | Total N = 314 | Unfavorable N = 80 | Favorable N = 234 | | | | |
| Male | 205 (65) | 60 (29.3) | 145 (70.7) | Reference | | | |
| Female | 109 (35) | 20 (18.3) | 89 (81.7) | 0.50 (0.3-1.0) | 0.040 | 1.8 (1.0-3.3) | 0.040 |
| Age | | | | | | | |
| 15 −24 | 97 (31) | 26 (26.8) | 71 (73.2) | Reference | | Reference | |
| 25-34 | 122 (39) | 33 (27) | 89 (73) | 0.80 (0.5-1.4) | | 1.0 (0.2-5) | 0.940 |
| 35-44 | 48 (15) | 13 (27) | 35 (72) | 1.00 (0.5-2.0) | 0.940 | 0.9 (0.2-4.8) | 0.930 |
| 45-54 | 29 (9) | 4 (14) | 25 (75) | 2.30 (0.8 −6.7) | 0.140 | 0.4 (0.1-2.7) | 0.380 |
| >55 | 18 (6) | 4 (27) | 14 (73) | 1.20 (0.4-3.8) | 0.740 | 0.7 (0.1-5.2) | 0.810 |
| HIV status | | | | | | | |
| HIV-negative | 283 (90) | 70 (26) | 213 (74) | Reference | | Reference | |
| HIV-positive | 31 (9.8) | 10 (23) | 21 (77) | 1.40.(0.6-3.2) | 0.360 | 1.4 (0.6-3.2) | 0.360 |
| Drug-resistance type | | | | | | | |
| RR-TB | 301 (95.5) | 78 (26) | 223 (74) | Reference | | Reference | |
| MDR-TB | 13 (4.1) | 2 (15.4) | 11 (84.6) | 0.47 (0.1-2.2) | 0.340 | 0.5 (0.1-2.2) | 0.340 |
| Registration group | | | | | | | |
| New | 64 (20) | 11 (17) | 53 (83) | Reference | | Reference | |
| Relapse | 145 (46) | 41 (28) | 104 (72) | 1.20 (0.7-2) | 0.430 | 1.1 (0.6.-2.2) | 0.690 |
| Loss to follow-up | 42 (12) | 15 (36) | 27 (64) | 1.20 (0.6-2.6) | 0.600 | 1.2 (0.5-3) | 0.660 |
| Treatment failure | 63 (21) | 13 (21) | 50 (79) | 0.60 (0.3-1.3) | 0.200 | 0.7 (0.3-1.6) | 0.440 |
| MDR-TB treatment | | | | | | | |
| Short-term | 60 (20) | 14 (23) | 46 (77) | Reference | | Reference | |
| Long-term | 254 (77) | 66 (26) | 188 (74) | 1.00 (0.6-1.9) | 0.920 | 1.0 (0.6-1.9) | 0.920 |
| Culture reversion | | | | | | | |
| Culture remained negative | 309 (98.4) | 75 (24) | 234 (76) | Reference | | Reference | |
| Culture reversed | 5 (1.6) | 5 (100) | 0.0 (0.0) | 0.05 (0.0-0.4) | 0.001 | 0.1 (0-0.4) | 0.001 |

Consequently, the current study finding complies with the World Health Organization's recommendation that the initial median time to sputum culture conversion to multidrug-resistant tuberculosis would be less than four months for a favorable treatment outcome prognosis [6].

In the current study, the cumulative probabilities of survival or sputum culture non-conversion at the end of the second, third, fourth, and sixth months were 0.90, 0.55, 0.22, and 0.15, respectively. However, in the previous study, the cumulative probabilities of survival in the first, second, and fourth months were 0.89, 0.56, and 0.19, respectively [20].

**Table 6. Logistic regression model selection using the different patient registration groups.**

| Model | Predictors | Log likelihood | LR X² (df) | p. value | Pseudo (R²) | AIC |
|-------|-----------|----------------|------------|----------|-------------|-----|
| Model 1 | New | −125.0 | 0.53 | 0.460 | 0.002 | 254.3 |
| Model 2 | New+ relapse | −124.8 | 1.29 | 0.526 | 0.001 | 255.5 |
| Model 3 | New+ relapse+ loss to follow-up | −124.4 | 2.10 | 0.724 | 0.008 | 258.7 |
| Model 4 | New +relapse+ loss to follow-up +treatment failure | −121.8 | 7.10 | 0.214 | 0.028 | 255.7 |

**Table 7. The association among patient registration group and treatment abandonment in the study participants (N.B. of the total 346 study participants, treatment outcome evaluated for 314 patients).**

| Registration group | N (%) | The current treatment abandonment N (%) | | COR | p-value | AOR | p-value |
|---------------------|-------|----------------|----------------|-----|---------|-----|---------|
| | | Yes, N = 52 | No, N = 262 | | | | |
| New | 64 (20.4) | 12 (18.7) | 52 (81.3) | Reference | | Reference | |
| Relapse | 145 (46.2) | 26 (18) | 119 (82) | 1.1 (0.6-2.2) | 0.67 | 1.2 (0.6-2.7) | 0.62 |
| Loss to follow-up | 38 (12) | 8 (21) | 30 (79) | 1.1 (0.4-2.7) | 0.91 | 1.2 (0.4-3.2) | 0.69 |
| Treatment failure | 67 (21.3) | 6 (9) | 61 (91) | 0.3 (0.1-1) | 0.04 | 2.4 (0.8-7) | 0.12 |

In the present study, the likelihood of sputum culture conversion rate was lower in the second and fourth months than in previous research. Suggesting that patients remained infectious for an extended period compared to earlier findings [20].

Unlike a previous study, which reported the association between delayed culture conversion and older age, the present study found no association between culture conversion and age groups [25].

The heterogeneity in findings across different regions of Ethiopia may be attributed to variations in patient characteristics, programmatic implementation of RR/MDR-TB treatment, diagnostic capacity, adherence and support, and timing of culture monitoring.

In the current study, there was no association between HIV status and time to culture conversion. This finding is consistent with some previous studies [9].

The absence of such an association may indicate that the treatment center managed the patients effectively and equitably, regardless of HIV status. However, other studies have reported earlier culture conversion among people living with HIV compared to HIV-negative individuals [13, 17, 22, 25].

A possible explanation for the early culture conversion among HIV positive patients in the previous studies might be due to the non-cavitary TB disease, which led to the low bacilli load, resulting in the culture negativity.

In the current study, there was no significant association between the type of drug resistance and time to culture conversion. The current finding was consistent with the previous study, which reported the absence of an association with any drug resistance type [22].

The current study revealed that patients with a history of prior loss to follow-up experienced an approximately fivefold delay in culture conversion, while relapse cases had a twofold delay, contrary to previous research [6].

A plausible explanation for the delayed culture conversion observed among patients who were lost to follow-up is suboptimal treatment adherence. Inadequate adherence—such as missed doses, irregular dosing schedules, or premature discontinuation of therapy—may compromise treatment efficacy and contribute to delayed microbiological response. Moreover, poor adherence can facilitate the emergence of additional drug resistance, further impeding culture conversion.

In both loss-to-follow-up and relapse cases, these resistant bacteria grow more slowly and are harder to kill, even when the patient restarts treatment.

Sometimes, tests may not detect all the resistance, so the treatment given might not be fully effective.

As a result of the above conditions, the sputum culture conversion delays, because the medicine doesn't work as well against the resistant bacteria.

Moreover, there was no significant association between the treatment regimen and time to culture conversion in the present study. In contrast, a previous study reported the association between kanamycin-containing treatment regimens and a delayed time to culture conversion [14].

It requires further investigation to generate more evidence and to determine the most effective treatment regimen for the patients.

In this study, we assessed 314 patients to determine their treatment outcomes. The overall treatment success rate was 234 (67.6%), comprising 177 patients (51.2%) who were cured and 57 (16.5%) who completed treatment successfully. Conversely, 80 (23%) patients experienced unfavorable outcomes, including 23 (6.6%) who died, 5 (1.4%) who experienced treatment failure, and 52 (15%) who were lost to follow-up.

The treatment success rate was lower in the current study, compared to the previous study's findings of 81% [26].

In addition, the current study demonstrated a lower rate of unsuccessful treatment outcomes compared with another previous study finding, which reported 4/189 (2%) and 32/189 (13%), death and treatment failure rates, respectively [27].

Unlike a previous study that reported unfavorable treatment outcomes, including deaths and loss to follow-up rates of 4.4% and 8.8%, respectively, this current study revealed markedly higher rates [28].

The higher rate of loss to follow-up observed among the study participants may be due to the patients' economic challenges or insufficient patient support. In addition, the adverse side effects of the medications might have discouraged the participants from continuing their treatment.

The high rate of loss to follow-up in another way contributed to treatment failure, relapse, and death, while the economic factor remained the underlying cause for the unfavorable treatment outcomes.

In the current study, female patients had 1.8 times higher odds of achieving a favorable treatment outcome, in contrast to the previous study's finding [28].

It demands further investigation to identify factors contributing to a favorable treatment outcome among the female sex groups.

On the other hand, out of the 302 patients who achieved culture conversion, 5 (1.66%) experienced culture reversion. However, the culture reversion rate was lower compared to the previous study, which had reported 54/1286 (4.2%) [9].

The study findings indicated that patients experiencing culture reversion had a 95% likelihood of facing an unfavorable treatment outcome, aligning with previous research [9].

However, this finding should be interpreted with caution, as it may be influenced by the small number of patients who experienced culture reversion.

## Conclusions

The majority of the patients achieved culture conversion within three months. However, patients with a history of loss to follow-up and relapse experienced delayed culture conversion. In addition, patients who had a culture reversion had worse treatment outcomes.

The death rate was also higher among the study participants.

These findings highlight the substantial need for sustainable monitoring to improve patient adherence.

## Recommendations

Maintain close monitoring of the patients, for those who experience delayed culture conversion, perform and repeat drug susceptibility testing to identify any additional drug resistance early.

Provide intensive health education and counseling on the disease and treatment adherence to the patients.

Provide sufficient financial support to cover food and transportation costs, to reduce unfavorable income-related treatment outcomes, including the high rate of loss to follow-up.

Further research is needed to investigate variables outside this study's scope and to yield more comprehensive findings.

### Limitations of the study

This study has several limitations, primarily related to its retrospective follow-up design. First, incomplete data in medical records posed a significant challenge. Key patient information—including marital status, educational level, religion, baseline alcohol use, smoking status, khat chewing habits, liver function tests, renal function tests, and CD4 counts for participants living with HIV—was not consistently available.

Additionally, not all patients underwent regular monthly sputum culture examinations. Incorporating these missing data points would have strengthened the evidence and reliability of the study findings.

If we had incorporated the additional data into our study, it would have strengthened the evidence presented.

The relatively small number of MDR-TB cases compared with RR-TB cases reduced the statistical power of the study, limiting the robustness and generalizability of the results.

Moreover, differences in the number of cases between the short-term and long-term treatment groups further constrained direct comparability between these groups.

To partially address these limitations, we applied censoring in the statistical analysis of time until sputum culture conversion. Nonetheless, these challenges highlight the need for additional prospective studies with more complete data collection to confirm and expand upon the findings reported here.

### Supporting information

**S1 File. Data collection form.**
(PDF)

### Acknowledgments

We would like to express our sincere gratitude to the Hawassa University School of Public Health for the guidance and professional advice provided throughout this work.

We also extend our appreciation to the leadership and staff of the Sidama Regional State Health Bureau, Sidama Public Health Institute, and Yirgalem General Hospital for their valuable support.

Our heartfelt thanks go to Mr. Adato Adela, Regional Laboratory Director, and Mr. Erdachew Ambaye, Quality Assurance Laboratory Team Leader, for their continuous guidance and encouragement. We further extend our sincere appreciation to the staff of the Yirgalem MDR-TB Treatment Center—especially Mr. Dagim Abebayehu, Dr. Wondimu Kasa, Mr. Abinet Alemayehu, and Mrs. Dinknesh Argaw—for their unreserved assistance during the research data collection process.

### Author contributions

**Conceptualization:** Wolde Abreham Geda.

**Data curation:** Wolde Abreham Geda, Kebede Tefera Betru, Tarekegn Solomon, Solomon Daniel.

**Formal analysis:** Wolde Abreham Geda, Kebede Tefera Betru, Tarekegn Solomon, Solomon Daniel.

**Funding acquisition:** Wolde Abreham Geda.

**Investigation:** Wolde Abreham Geda, Kebede Tefera Betru, Tarekegn Solomon, Solomon Daniel.

**Methodology:** Wolde Abreham Geda, Kebede Tefera Betru, Tarekegn Solomon, Solomon Daniel.

**Project administration:** Wolde Abreham Geda.

**Resources:** Wolde Abreham Geda.

**Software:** Wolde Abreham Geda.

**Supervision:** Wolde Abreham Geda, Kebede Tefera Betru, Tarekegn Solomon.

**Validation:** Wolde Abreham Geda, Kebede Tefera Betru, Tarekegn Solomon, Solomon Daniel.

**Visualization:** Wolde Abreham Geda, Kebede Tefera Betru, Tarekegn Solomon.

**Writing – original draft:** Wolde Abreham Geda, Kebede Tefera Betru, Tarekegn Solomon.

**Writing – review & editing:** Wolde Abreham Geda, Kebede Tefera Betru, Tarekegn Solomon, Solomon Daniel.

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
