## [Decision Letter · Decision Letter 0]

15 Sep 2025

Dear Dr. Geda,

Thank you for submitting your manuscript to PLOS ONE. After careful consideration, we feel that it has merit but does not fully meet PLOS ONE’s publication criteria as it currently stands. Therefore, we invite you to submit a revised version of the manuscript that addresses the points raised during the review process.

Please submit your revised manuscript by  Oct 30 2025 11:59PM. If you will need more time than this to complete your revisions, please reply to this message or contact the journal office at plosone@plos.org . . A rebuttal letter that responds to each point raised by the academic editor and reviewer(s). You should upload this letter as a separate file labeled 'Response to Reviewers'.A marked-up copy of your manuscript that highlights changes made to the original version. You should upload this as a separate file labeled 'Revised Manuscript with Track Changes'.An unmarked version of your revised paper without tracked changes. You should upload this as a separate file labeled 'Manuscript'.

We look forward to receiving your revised manuscript.

Kind regards,

Pedro Eduardo Almeida da Silva

Academic Editor

PLOS ONE

Journal Requirements:

Reviewers' comments:

Reviewer's Responses to Questions

**Comments to the Author**

1. Is the manuscript technically sound, and do the data support the conclusions?

Reviewer #1: Partly

Reviewer #2: No

2. Has the statistical analysis been performed appropriately and rigorously?

Reviewer #1: No

Reviewer #2: I Don't Know

3. Have the authors made all data underlying the findings in their manuscript fully available?

Reviewer #1: Yes

Reviewer #2: No

4. Is the manuscript presented in an intelligible fashion and written in standard English?

Reviewer #1: No

Reviewer #2: No

Reviewer #1: Critical comments

1- Review the sample size calculation and sampling strategy. The description is confusing. The authors calculated a sample size and designed a sampling plan, but in the end included all patients without reaching the calculated N. The authors must clearly state what was actually done, i.e., the inclusion of all eligible individuals, and, if appropriate, calculate the statistical power of the achieved sample based on the observed differences.

2- Even though this is a retrospective study, patients were followed in some manner. The authors must specify how follow-up was conducted and for how long.

3- Provide detailed definitions of all evaluated outcomes.

4- The study included multiple treatment records with different timeframes, resulting in unbalanced populations across treatment groups. The authors need to explain how they addressed this source of bias.

5- The inclusion of relapse cases introduces an established risk factor for treatment abandonment. The authors must clarify how they accounted for this in the analysis.

6- Provide a clear and detailed description of the statistical analyses. Specify which measures were calculated, which tests were used and for which variables, how assumptions were assessed, and how results were interpreted.

7- Present a more detailed description of the Cox regression models. The authors must specify dependent and independent variables, adjustments performed, how these adjustments were evaluated, and how results were reported.

8- Revise the discussion to synthesize the main findings and critically interpret them in light of the study’s focus. The reported reduction in culture conversion time compared with other studies requires explanation. Could bias have contributed to this finding? What are the clinical and practical implications?

9- Discuss the implications of the treatment outcome findings for the management of patients with resistant or multidrug-resistant tuberculosis.

10- Explicitly state how the described limitations affect the interpretation of results and indicate how the authors attempted to mitigate them.

11- Clearly and consistently state the study objective. As currently written, the objective remains ambiguous.

Major comments

1- Revise the “source and study population” section and the inclusion/exclusion criteria. These subsections are redundant.

2- Reconsider the exclusion of individuals under 15 years of age. It is redundant to exclude a population that was not initially included in the study.

3- Revise the ethics and informed consent subsections. These are redundant.

4- Strengthen the description of the study population. The current description is weak and insufficient to understand the characteristics of the participants.

5- Figure 2 requires better presentation in the Results section. The authors should specify which treatments were associated with shorter culture conversion times.

6- Revise the Conclusion. It should directly answer the research question and restate the study objective. Additional sentences belong in the Discussion section.

Minor comments

1- Figures require more detailed data presentation to facilitate interpretation. Improve figure quality.

2- Revise the manuscript thoroughly to improve readability. Issues of textual cohesion and sentence connection reduce fluency.

Reviewer #2: PLOS ONE

Time-to-sputum culture conversion, treatment outcomes, and associated factors among multidrug-resistant tuberculosis patients in the Sidama region, Ethiopia: A retrospective follow-up study

The authors have conducted a retrospective follow-up study on 346 patients aimed to determine the time-to-culture conversion, treatment outcomes, and associated factors among multidrug-resistant tuberculosis in the Sidama Region, Ethiopia from 2013-2024.

The authors conclude that most of the patients (87.3%) achieved culture conversion within three months. The patients with a history of retreatment faced worse outcome, pointing to serious treatment follow-up to the patients.

Such studies are very important to evaluate the treatment and factors associated with failure to improve outcomes of patients with MDR-TB. Still, the manuscript seems preliminary and has several limitations and should be substantially revised to make their data more accessible for the reader.

The Introduction is partly repetitive on the importance of the study. Still, there is no background or references on what is already known and previously published about time-to-culture conversion, treatment outcomes, and associated factors (only in Discussion). Also, the 6 months BPaLm regimen is not mentioned (Line 72: “MDR-TB treatment takes longer, 18-20 months, and it is very costly”).

In the Methods the definitions of various parameters should be better explained. Ex. Line 171: short-term, long-term and standardized treatment should be explained since this could have relevance for side effects and outcome. I am not qualified to evaluate all the statistical methods used but the data collection and statistical method are not very well explained. Also, the implications of the sample size calculations are not clear as the authors did not use these data as the numbers of patients available were much fewer.

The discussion is partly repetitive of results and just state where their findings were similar or different from other studies without discussing or giving possible explanations for their findings. The Conclusion is also a repetition from Results and Discussion and Limitations were deficient.

The manuscript also needs substantial update and general editing concerning language, fonts (Tables), typing errors including correct and not repetitive use of abrev. as ex. Tuberculosis (TB) and Multidrug-resistant/ Rifampicin mono-resistant- TB (MDR/TB) that have been used several times.

Line 34 needs ref. The WHO 2024 report will most likely provide data for 2023 including for Ethiopia.

Line 102: such a detailed description of the country is probably not needed.

Line 121: It is not clear concerning the time period when the patient data were collected: “ The patients were registered from January 1, 2013, to June 30, 2024, from the Sidama region of Ethiopia at the Yirgalem treatment initiation center. The researchers accessed the patient data anonymously, using their medical and multidrug-resistant TB record number, from April 1 to December 31, 2024”.

Line 134: “…. patients with culture-negative results at baseline, and patients diagnosed with MDR/RR-TB who did not begin anti-TB treatment..” An explanation on who these patients are and why not treated would be interesting.

Line 142: not clear if the same calculations were performed for all outcomes? Also, since the largest sample size for a single-arm cohort study was = 1977, but the total eligible RR/MDR-TB population in the region during the study period was finally 346, are the study powered to answer the research questions and are the sample size calculations of any value?

Line 205: Participant agreement is already stated in the section above.

Line 209: Write more precise concerning which patients included “A total of 420 patients with Multidrug-resistant tuberculosis enrolled over an 11-year follow-up”. This statement disagrees with Table 3 were also RR and INH resistance are included.

Line 262: People who registered with a drug-resistance type of MDR-TB were associated with early time to-sputum culture conversion with aHR 4.7 (1-21), P = 0.041 compared with the patients with a resistance type of RR-TB, while a treatment regimen of long-term MDR-TB treatment was significantly associated with an aHR of 10.1 (2.2-45), P = 0.003 compared with the short-term treatment. This statement is not so clear formulated and not as expected that MDR-TB should do better than rifampicin mono resistance? Still, the authors just state this and provides no possible explanation for their findings in the discussion (Line 351).

Line 280: “The overall treatment success rate was 234 (67.6%). The cure and treatment completion rates were 177 (51.2%) and 57 (16.5%), respectively”. How could treatment success be higher than cure rate? How was cure discriminated from treatment completion?

Line 294: Factors associated with treatment outcomes. HIV were associated with unfavourable treatment outcomes. If possible CD4 count and of PLWH were on ART or not would be valuable to evaluate the impact of HIV infection

**Do you want your identity to be public for this peer review?** For information about this choice, including consent withdrawal, please see our For information about this choice, including consent withdrawal, please see our Privacy Policy .

Reviewer #1: **Yes:** João Paulo ColaJoão Paulo Cola

Reviewer #2: No

While revising your submission, please upload your figure files to the Preflight Analysis and Conversion Engine (PACE) digital diagnostic tool, https://pacev2.apexcovantage.com/ . PACE helps ensure that figures meet PLOS requirements. To use PACE, you must first register as a user. Registration is free. Then, login and navigate to the UPLOAD tab, where you will find detailed instructions on how to use the tool. If you encounter any issues or have any questions when using PACE, please email PLOS at . PACE helps ensure that figures meet PLOS requirements. To use PACE, you must first register as a user. Registration is free. Then, login and navigate to the UPLOAD tab, where you will find detailed instructions on how to use the tool. If you encounter any issues or have any questions when using PACE, please email PLOS at figures@plos.org . Please note that Supporting Information files do not need this step.. Please note that Supporting Information files do not need this step.

---

## [Author Response · Author response to Decision Letter 1]

20 Nov 2025

I attached it in the 'response to the reviewers' file.

---

## [Decision Letter · Decision Letter 1]

3 Feb 2026

Dear Dr. Geda,

Thank you for submitting your manuscript to PLOS ONE. After careful consideration, we feel that it has merit but does not fully meet PLOS ONE’s publication criteria as it currently stands. Therefore, we invite you to submit a revised version of the manuscript that addresses the points raised during the review process.

Please submit your revised manuscript by Mar 20 2026 11:59PM. If you will need significantly more time to complete your revisions, please reply to this message or contact the journal office at plosone@plos.org . . A letter that responds to each point raised by the academic editor and reviewer(s). You should upload this letter as a separate file labeled 'Response to Reviewers'.A marked-up copy of your manuscript that highlights changes made to the original version. You should upload this as a separate file labeled 'Revised Manuscript with Track Changes'.An unmarked version of your revised paper without tracked changes. You should upload this as a separate file labeled 'Manuscript'.

We look forward to receiving your revised manuscript.

Kind regards,

Frederick Quinn

Academic Editor

PLOS One

**Journal Requirements:**

Reviewers' comments:

Reviewer's Responses to Questions

**Comments to the Author**

Reviewer #1: All comments have been addressed

Reviewer #2: (No Response)

2. Is the manuscript technically sound, and do the data support the conclusions?

Reviewer #1: Yes

Reviewer #2: No

3. Has the statistical analysis been performed appropriately and rigorously?

Reviewer #1: Yes

Reviewer #2: I Don't Know

4. Have the authors made all data underlying the findings in their manuscript fully available?

Reviewer #1: Yes

Reviewer #2: Yes

5. Is the manuscript presented in an intelligible fashion and written in standard English?

Reviewer #1: Yes

Reviewer #2: No

**Reviewer #1:** The authors implemented the revisions addressing the points raised in the review; however, they did not submit a response letter. The manuscript still requires minor editing to standardize decimal places in the results (particularly for relative frequencies) and to correct the ordering/numbering of the figures, as some figures are not cited in the text and/or are mislabeled.The authors implemented the revisions addressing the points raised in the review; however, they did not submit a response letter. The manuscript still requires minor editing to standardize decimal places in the results (particularly for relative frequencies) and to correct the ordering/numbering of the figures, as some figures are not cited in the text and/or are mislabeled.

**Reviewer #2:**  The authors have improved the paper but is still needs substantiell revision in content and editing. The main objection is that the authors aim to present data from MDR-TB patients whereas only 5% or less of the cohort are MDR-TB, Thus, the title “Time-to-sputum culture conversion, treatment outcomes, and associated factors among multidrug-resistant tuberculosis patients….” as misleading as most of the data presented are from RR patients and this is not discussed as a limitation. Also, they present data partly diverging from the first version and it is unclear if this is a result of re-analyses correcting previous results or new statistical methods. The authors have improved the paper but is still needs substantiell revision in content and editing. The main objection is that the authors aim to present data from MDR-TB patients whereas only 5% or less of the cohort are MDR-TB, Thus, the title “Time-to-sputum culture conversion, treatment outcomes, and associated factors among multidrug-resistant tuberculosis patients….” as misleading as most of the data presented are from RR patients and this is not discussed as a limitation. Also, they present data partly diverging from the first version and it is unclear if this is a result of re-analyses correcting previous results or new statistical methods.

**Do you want your identity to be public for this peer review?** For information about this choice, including consent withdrawal, please see our For information about this choice, including consent withdrawal, please see our Privacy Policy .

Reviewer #1: **Yes:** João Paulo ColaJoão Paulo Cola

Reviewer #2: **Yes:** Anne Ma Dyrhol-RiiseAnne Ma Dyrhol-Riise

---

## [Author Response · Author response to Decision Letter 2]

4 Mar 2026

To

The Editor and Reviewers

PLOS ONE

Date: March 4, 2026

Dear Editor,

Subject: Response to Reviewers’ Comments for Manuscript ID: PONE-D-25-32357

We thank you and both reviewers for the careful evaluation of our manuscript and for the constructive comments provided. We have revised the manuscript accordingly and believe that these changes have substantially improved its clarity, rigor, and alignment between the study objectives and the data presented.

Response to Reviewer #1

We acknowledge the comments regarding the absence of a response letter and the minor editorial issues noted in the original submission. In the revised version, we have included a comprehensive response letter addressing all reviewer comments. In addition, we thoroughly edited the manuscript to standardize decimal places throughout the Results section, particularly for relative frequencies. We also corrected the ordering, numbering, and labeling of all figures and ensured that each figure is appropriately cited and referenced in the text.

Response to Reviewer #2

We appreciate the reviewer’s detailed feedback regarding the scope, content, and interpretation of the data, which prompted substantial revisions to the manuscript. First, we revised the title and related sections to better reflect the composition of the study cohort and the data presented. We now explicitly acknowledge the imbalance between MDR-TB and RR-TB patients and clearly discuss this issue as a limitation.

Second, we confirm that the dataset used in the revised manuscript is identical to that of the original submission. In response to the reviewer’s suggestions, we conducted new statistical analyses using additional and more appropriate models. These revised models yielded different effect estimates for some variables, reflecting improvements in the suitability of the statistical methods rather than changes in the underlying data. Consequently, we updated the Results, Discussion, and Limitations sections to ensure consistency with the revised analyses and to provide a clearer interpretation of the findings in light of these methodological improvements.

For ease of review, we provide a detailed, point-by-point response outlining how each concern has been addressed below.

We greatly appreciate the time and effort devoted to reviewing our work and the constructive engagement throughout the process.

Sincerely,

Wolde Abreham Geda

---

## [Decision Letter · Decision Letter 2]

17 Mar 2026

Time to sputum culture conversion, treatment outcomes, and associated factors among rifampicin-resistant or multidrug-resistant tuberculosis patients in the Sidama region, Ethiopia: A retrospective follow-up study

PONE-D-25-32357R2

Dear Dr. Geda,

We’re pleased to inform you that your manuscript has been judged scientifically suitable for publication and will be formally accepted for publication once it meets all outstanding technical requirements.

Kind regards,

Frederick Quinn

Academic Editor

PLOS One

Additional Editor Comments (optional):

Reviewers' comments:

Reviewer's Responses to Questions

**Comments to the Author**

Reviewer #1: All comments have been addressed

2. Is the manuscript technically sound, and do the data support the conclusions?

Reviewer #1: Yes

3. Has the statistical analysis been performed appropriately and rigorously?

Reviewer #1: Yes

4. Have the authors made all data underlying the findings in their manuscript fully available?

Reviewer #1: Yes

5. Is the manuscript presented in an intelligible fashion and written in standard English?

Reviewer #1: Yes

Reviewer #1: All concerns and questions previously raised during the review process have been adequately addressed by the authors. The revisions provided satisfactorily clarify the issues identified in the earlier evaluation, and the manuscript has been appropriately improved. The responses are consistent with the reviewers’ comments, and the modifications made to the text are sufficient to resolve the points previously highlighted. Therefore, no further clarification is required at this stage.

**Do you want your identity to be public for this peer review?** For information about this choice, including consent withdrawal, please see our For information about this choice, including consent withdrawal, please see our Privacy Policy .

Reviewer #1: No

---

## [Editor Report · Acceptance letter]

PONE-D-25-32357R2

PLOS One

Dear Dr. Geda,

I'm pleased to inform you that your manuscript has been deemed suitable for publication in PLOS One. Congratulations! Your manuscript is now being handed over to our production team.

Kind regards,

on behalf of

Dr. Frederick Quinn

Academic Editor

PLOS One